# Validating the Contribution of Nature-Based Farming Solutions (NBFS) to Agrobiodiversity Values through a Multi-Scale Landscape Approach

Ilda Vagge * and Gemma Chiaffarelli

Department of Agricultural and Environmental Sciences, University of Milan, Via Celoria 2, I-20133 Milan, Italy
* Correspondence: ilda.vagge@unimi.it

**Abstract:** Nature-Based Farming Solutions (NBFS) are envisaged practices that still strongly demand further context-specific scientific validation for their viable deployment at the local scale. In this context, our study deals with the test of a multi-scale system of landscape ecology indicators, interpreted as surrogates for the accounting of the contributions of NBFS to agrobiodiversity values and to the consequent environmental stability and resilience capacities of agroecosystems, recognized as pivotal for facing the ongoing climate change challenges. We here present the preliminary results obtained in a first pilot case study (Po Plain context). Landscape ecology analyses were undertaken at extra-local, local, and farm scales (with different levels of analytical detail), comparing the pilot farm to the surrounding conventionally managed context. A set of structural and functional indicators were tested, allowing a preliminary screening of the most suitable ones (good sensitivity to treatment changes, informative potential). Results suggested a multi-faceted positive contribution given by NBFS implementation and were the basis for orienting further NBFS implementation strategies based on vulnerability and resilience properties analysis. Further investigations are envisaged on wider datasets coming from other pilot case studies belonging to similar pedo-climatic conditions, in order to improve the informative potential of the here presented methodology.

**Keywords:** Nature-Based Farming Solutions; landscape ecology; agrobiodiversity; multi-scale approach

## 1. Introduction

Nature-Based Farming Solutions (NBFS) (here intended as landscape feature creation and maintenance across the rural landscape, agroforestry practices, and crop diversification) are oriented towards the reintegration, within the agricultural systems, of their undermined ecosystem functions [1–8]. Thus, they are aimed at enhancing the environmental stability, resistance, and resilience capacity [9] of agroecosystems and their consequent climate change adaptation and mitigation suitability [10–21]. The latest European and worldwide agri-environmental policies and sustainable development strategies are recognizing such integrated approaches as pivotal [22–30].

Despite these premises, consistent and successful integration of NBFS among real farming systems has been delayed despite its claimed urgency [10,31–33]. European and national public subsidies are supporting such approaches through diversified policy tools [5,13], which are mainly derived from practice-based evaluation approaches. Result-oriented approaches would overcome the limits of linking subsidies to pre-established specific practices. Indeed, they would allow the support of a farm management model as a whole on the basis of its recognised positive influence on significant agri-environmental indicators [13,32,34], accounting for an influence on ecological functions. Nonetheless, result-based approaches still need to be fully developed (first of all, by overcoming cost-efficiency performance limits), calibrated, and validated among specific pedo-climatic contexts in order to be successfully integrated into the different levels of policy and accreditation tools [11,32,35–41]. Hybrid assessment schemes (both practice- and result-based) are often

the most viable solution [35]. Different experiences already exist with integrated, result-based, and hybrid agrobiodiversity indicator monitoring systems [40,42–49]. Multi-scale analytical approaches are envisaged for such kinds of assessments, as field-scale biodiversity is intrinsically influenced by factors working on different spatial scales [15,50,51]. Nonetheless, general concepts, theories, and models linking biodiversity to landscape scale parameters only allow us to use them as a surrogate of agrobiodiversity values [52], while their use as a correlate to biodiversity values still needs context specific validation [49,52]. Given this, landscape ecology tools [53–59] validated for specific contexts offer a viable double contribution, both accounting for structural (in link with management options) and functional traits (in link with results derived from management options).

In continuity with this framework, our project deals with a context-specific test, calibration, and validation of a multi-scale landscape ecology indicators system through the study of organic farming systems adopting agroecological practices that can be ascribed to the NBFS approach [5,60]. The final aim is the selection of a viable set of indicators effective in synthesizing the contributions of NBFS to overall farm agrobiodiversity values and, in parallel, the orientation of further NBFS implementation strategies [61,62]. The contexts under study are located among different districts of the Po Plain system (North of Italy), sharing similar (but not identical) climatic conditions, and representing different pedological case histories of this alluvial macro-context.

In this paper, we present preliminary results from a first pilot case study compared to its surrounding conventionally managed agricultural situation. These results, which already highlighted positive trends on the studied pilot farm, are interpreted here as a surrogate representation of agrobiodiversity values from a hybrid practice and result-based perspective. The in-plan comparison of the here-presented results with floristic-vegetational indicators [63–67] will allow for their use as a correlate to biodiversity values [52]. This will further extend their informative potential on the ecological patterns and trends resulting from farmland agroecological management through the NBFS [6,42,43,68–74], while also orienting its strategic redesign through time [3,4,61,62,75–80].

## 2. Materials and Methods

### 2.1. The Adopted Landscape Ecology Analytical Approach

Landscape ecological structural and functional traits were studied following a multi-scale approach (extra-local scale; local scale (*La*); and farm scale (*Fa*)) (Figure 1), making reference to the landscape ecology [53–58,81] and landscape bionomics methodologies [78,82,83]. The set of indicators to be analyzed was set up, including physiognomic-structural, functional, and dynamic information levels on the landscape system. Appendix A; Table A1 lists: (i) the set of indicators under study; (ii) the related formulas; and (iii) the main references for each indicator under study (Appendix A; Table A1). Potentially redundant variants of the same indicator categories were included. The aim was to: (i) assess their sensitiveness to the compared management options among the specific studied context; and (ii) consequently, identify a preliminary sub-set of independent (low correlation coefficients values) and highly sensitive indicators, that were viable for the project's purposes.

Extra-local and local scale boundaries were identified according to the landscape unit and ecotope concepts developed by Ingegnoli [78,82,84]. The ratio of local scale to extra-local scale surface resulted in about 1% (3.44 km$^2$ and 312.05 km$^2$, respectively). Farm scale boundaries were set coincident to farm's patches boundaries (0.12 km$^2$), even if discontinuous, in order to only take into account in-farm management options effects.

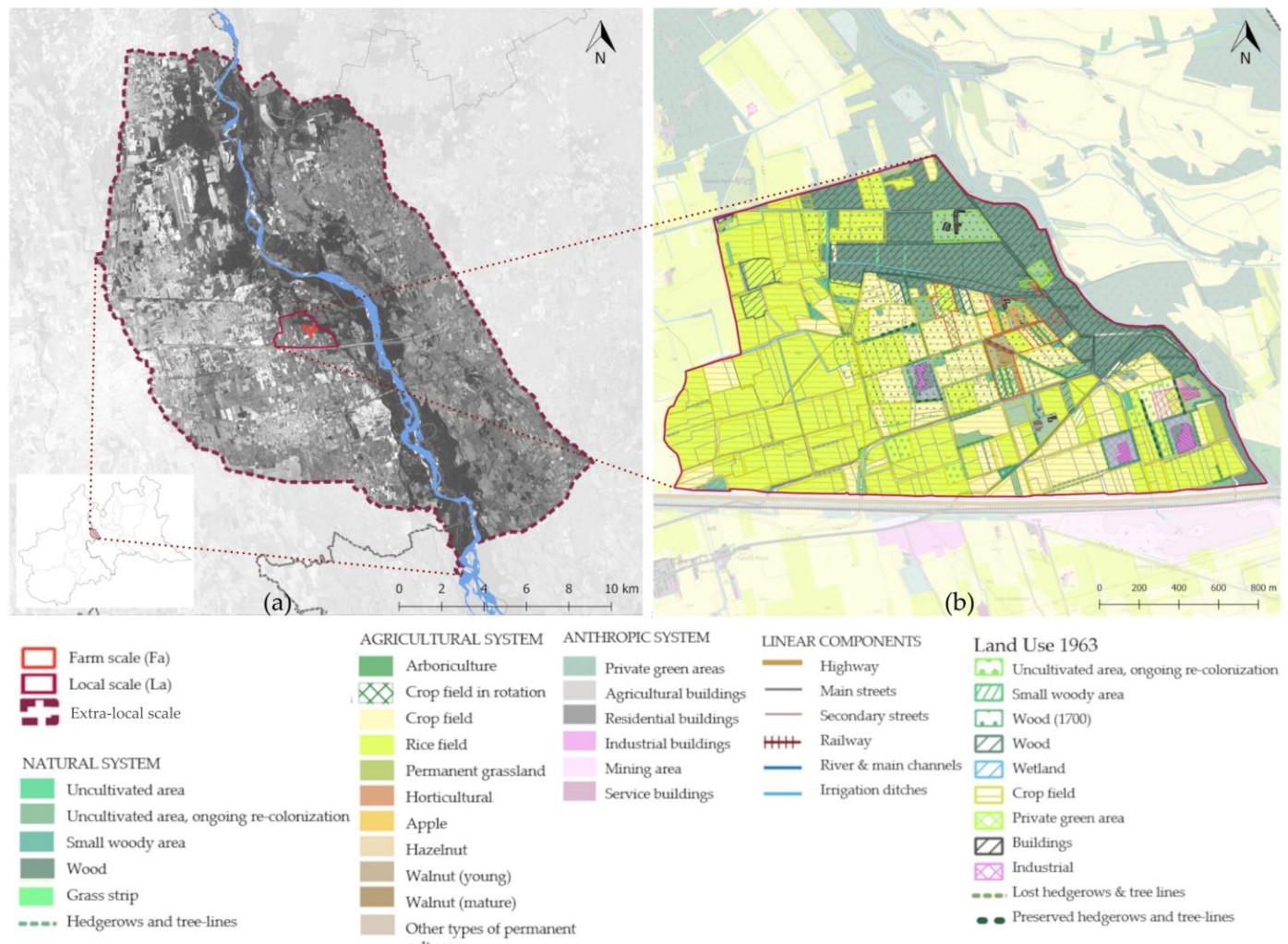

**Figure 1.** (**a**) Pilot farm location among the regional, extra-local, and local contexts, along the Ticino River basin; (**b**) land cover at local scale (LCP 2010, Piedmont region [85]), showing the current landscape patches and linear corridors (as used for landscape metrics computing), compared to historical traits.

Spatial-qualitative featuring of the extra-local and local scale analytical units was based on the following GIS information [85–87]: geomorphology, pedology, hydrology, phyto-climate, regional land cover, historical land use, vegetation, protected areas, regional ecological network, and other in-force land-planning tools. At an extra-local scale, this preliminary context informed an exploratory vulnerability and resilience (VR) analysis [88–92]. For local scales, GIS data were validated through coarse field surveys.

At *La* and *Fa* scales, current landscape mosaic structural elements (matrix, patches, and corridors) were mapped as vectors on QGIS software (the minimum patch size was set to 70 m²) and quantified [by number, relative surfaces, and perimeters cumulated for each land use significant category, clustered in 3 landscape sub-systems: natural (NAT), agricultural (AGR), and anthropic (ANT) (Figure 1)]. Preliminary framing analyses were carried out on landscape physiological apparatuses, whose spatial configuration and relative surfaces were assessed at the *La* scale, referring to the *eco-tissue model* [78,82,93]. Then, landscape ecology metrics (see Appendix A; Table A1) were calculated separately for each landscape sub-system and then jointly (total *La* and *Fa* area). This allowed a

comparison between the pilot farm and the surrounding conventional farms, through percentage gap evaluation, as shown below:

$$\text{GAP } [\%] = \frac{(X_{Fa} - X_{La})}{X_{La}},$$

with $[X_{Fa}]$; $[X_{La}]$ = each $X$ indicator value, respectively, at the *Fa* and *La* scales.

Another synthetic functional indicator, the Biological Territorial Capacity (BTC) [73,78,94], was computed and spatially represented at *La* and *Fa* scales using literature BTC values [78,94]. BTC evaluates the metastability of the ecological mosaic and is here considered useful for contextualizing, from an ecological functional point of view, the results coming from the set of indicators under study.

In order to take into account the influence of the ecological quality of links on effective connectivity and circuitry functions, a synthetic weighting system was newly tested, adapted from Fabbri [95], through the attribution of each *i*-link to one of 5 Ecological Quality Classes ($EQC_y$), based on quick field surveys and qualitative information (Table 1). Each *i*-link $EQC_y$ value ($EQC_{yi}$) was calculated as follows:

$$EQC_{yi} = \frac{Strat_i + Dev_i + Cont_i + Autoct_i}{4},$$

with $EQC_{yi} = [1\text{–}5]$.

**Table 1.** The four criteria used for links' Ecological Quality Classes ranking.

| Value | Stratification (Strat) | Development Degree (Dev) | Continuity (Cont) | Autochthonous Degree (Autoct) |
|---|---|---|---|---|
| 1 | No stratification | Low development | Low | Consistent allochthonous species |
| 2 | Low stratification | Low–medium development | Low–medium | |
| 3 | Mixed stratified and no stratification | Medium development | Medium | Autochthonous and allochthonous species |
| 4 | Stratified | Medium–well developed | Medium–good | |
| 5 | Highly stratified | Well developed | Good | Autochthonous species Dominant |

Connectivity and circuitry metrics values were compared between local and farm scales as well, in order to highlight the differences between the pilot farm landscape management options if compared to their surrounding conventionally managed context with respect to their contributions to biotic flux support across the landscape matrix.

All these analyses provided information for the resulting VR analysis at the local scale, which was conceived as a contextualizing precondition tool for interpreting landscape metrics, orienting the strategic agroecological farm management planning over time, and orienting the consequent in view of the ES assessment [88]. The basic data analyzed are contained in Tables S1–S10 in the Supplementary Materials.

### 2.2. The Pilot Case Study: Environmental Context

The pilot case study on which the set of indicators was tested is an organic farm located on fluvial and fluvioglacial Würm deposits among the western Po Plain basin (Piedmont Region), nearby the Ticino River protected ecological corridor, in an agricultural context dominated by intensive conventional rice and corn production (Figure 1). This small-size pilot farm (about 13 ha) is characterized by highly diversified annual productions, under 3 years rotations (rice, barley, corn, leguminous plants, mixed horticulture), and permanent fruit productions (walnut, nut, apple). Traditional management practices are adopted, compliant with organic agriculture requirements, including the use of landraces, avoiding

the use of chemical fertilizers and herbicides, recovering ancient small-sized plot texture, and the preservation and restoration of the ecological infrastructure through landscape features management (hedgerows, tree lines, riparian vegetation along irrigation ditches, wooded areas, and wooded patches).

Bioclimatically, the area belongs to the temperate oceanic (submediterranean) bioclimate, the upper meso-temperate thermotype, and the lower humid ombrotype [96,97]. The real potential vegetation is represented by the "Neutroacidophilous series of European oak and hornbeam in the lower western Po Valley" [98–100]. Historical land use traits (IGM 1963 [86]; and an *ancient XVIII century map by the Turin State Archives*) (Figure 1) highlight the long-term stability of wooded contexts, the contraction of some other wooded areas, and the progressive dismantling of the linear landscape features system. Current phytocoenoses are significantly influenced by synanthropic secondary dynamisms, especially the more unstable ones (wood margins, hedgerows, and young small wooded areas). The local flora, currently under study by the authors, is composed of 132 taxa (species and sub-species), belonging to 49 taxonomic families, mainly hemicryptophytes and therophytes and, secondarily, phanerophytes, and then geophytes, reflecting bioclimatic conditions and a variety of open and wooded habitats with different degrees of disturbance. Eurasian macrochorotypes predominate, as expected given the biogeographic context, and species with a wide distribution in relation to anthropization. This also results in a significant presence of allochthonous species; in fact, 27.48% of the total flora is represented by allochthonous species, of which 75% are invasive exotics, while the Italian national average is 19.49% [101].

## 3. Results and Discussion

### 3.1. Landscape Ecology Analyses Results

Extra-local spatial-qualitative VR assessment (Figure 2) helped in contextualizing the pivotal buffering and permeating role that the local scale context and the pilot farm area can play, being located at the interface between a wide, protected primary ecological corridor axis and the surrounding, more simplified Po Plain agricultural context. In such a context, rural landscape features and equipment can positively spread the biotic exchanges from the neighboring source areas, broadening their effects on west-east axes through the activation of new ecological corridors. In parallel, NBFS implementation can contribute to buffering the sink effects and pollutants' impacts coming from the agricultural matrix and major infrastructures affecting the ecological quality of the main fluvial ecological corridor.

The local scale preliminary functional variability revealed through the spatial-quantitative assessment of landscape physiological apparatuses (Figure 3a,b) highlighted a relatively good balance of stabilization and protective functions (19.8% and 5.5% of total surface, respectively). In contrast, connective functions appear to be underrepresented (1.6%), reflecting the bipolar configuration of the area, where areas with high metastability behavior (the northeastern wooded protected areas) are poorly permeated by the agricultural ones, and buffering functions between the two systems are underexpressed. The low-represented resilience functions (2.4%), performed by low-developed secondary recolonization contexts, reflect the medium disturbance level on semi-natural patches external to the more consistent stabilizing wooded patches and, in parallel, represent a potential driver of change, if properly managed. As a result, the current impact of agricultural management disturbances on floral-vegetational ecological and dynamic phytocoenoses of the agricultural system is expected, with a notable difference in wooded area traits.

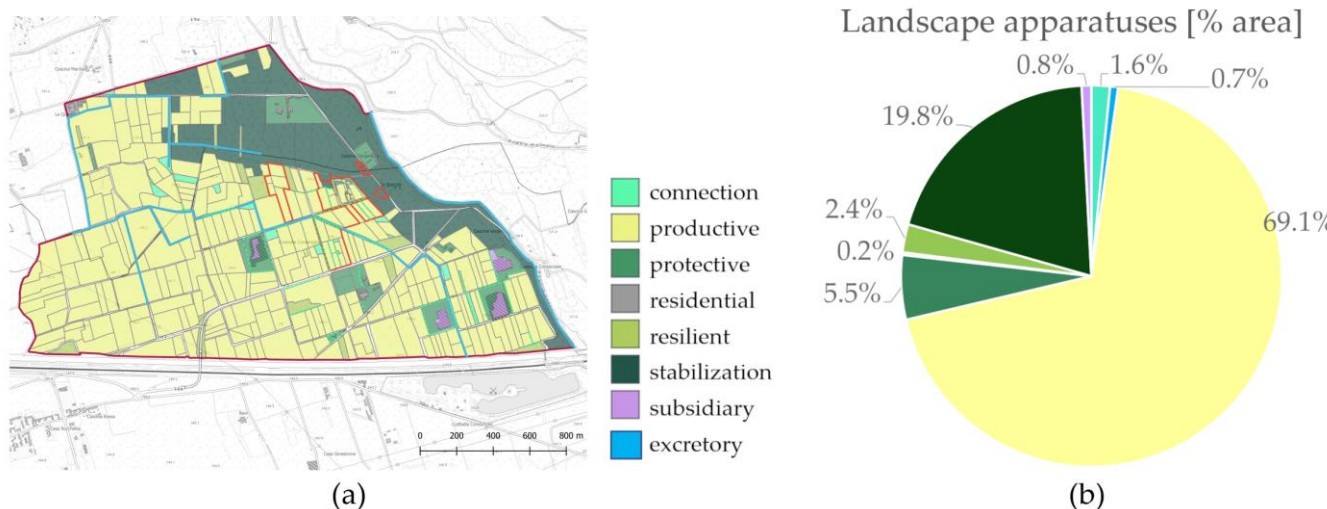

**Figure 2.** Extra-local scale VR analysis.

**Figure 3.** (**a**) Local-scale landscape apparatuses' spatial configuration. (**b**) Local scale landscape apparatuses percentage distribution ($area_i$/total area; with $area_i$ = total area of each landscape apparatus).

3.1.1. Landscape Structural Indicators Results

*Fa* represents 3.7% of the total *La* area, 6.8% of the total *La* perimeter, and 11.3% of the total *La* number of patches. This relation already suggests higher total shape complexity for *Fa* (a higher total perimeter/area ratio), with smaller patches.

Results on shape indices (Tables 2 and 3; Figure 4) highlighted the following traits (see Appendix A; Table A1 for details on the studied indicators):

- Positively lower MPS values for *Fa* (MPS equal to 0.43 ha; −19% for the AGR *Fa* sub-system), which reflect the positive effects related to small-sized agricultural patches among the pilot farm: higher landscape heterogeneity [102,103]; hedge microhabitat and niche diversification [104–106]; lower sink effects [54,107]; increased variability of micro-climatic conditions and provision of breeding sites [108,109]; and encouraging species diversification and higher avian diversity (a balance between generalist and specialist behaviors) [54,110–112];

- Relatively high MPS values for the *La* NAT sub-system (MPS equal to 1.12 ha) reflect a proper consistency of natural (woody) patches among *La*. This can be related to a positive influence on the properties of source areas: interior habitat conditions, higher specialist species population sizes, lowering their probability of local extinction, rising stability traits, and balancing the impacts coming from the neighboring agricultural matrix [13,54]. These traits underline the current pivotal role of the existing woody components and the potential that could be displayed by their preservation, proper management (improvement of their ecological quality), and extension;

- The MTX index showed, as expected: lower occurrence of natural sub-system components in *Fa* (−63%), which might be enhanced through the further integration of diffused natural and semi-natural components and in between productive fields [102,104,105]; higher relative occurrence in *Fa* of agricultural sub-system components (+22%); and good stability of the *La* agricultural matrix (absence of significantly spread impacts derived from non-compatible land uses) (69.6% of total surface), as it was already highlighted by landscape apparatus analysis (Figure 3a,b). This increases the potential for effectively correcting the current ecological trends toward higher quality values [13,113–115];

- Indices related to patch shape complexity (MPAR, SI, PFD, and their variants) showed positive results (significantly higher patch complexity) for *Fa* in their arithmetic mean variants, where the effect of the different *Fa* and *La* overall dimensions is kept under control (average single patch values): MPAR (+623% in TOT *Fa*), MSI (+201% in TOT *Fa*), and MPFD (+438% in TOT *Fa*). Thus, MPAR, MSI, and MPFD showed good sensitivity to *La* and *Fa* differences. They were also positively correlated (Table 3) and can be consequently considered redundant in the studied context (they capture similar qualities of spatial patterns); their potential interchange might be further tested among larger datasets. In contrast, their area-weighted mean variants (AWSI and AWPFD) showed negative results for *Fa*. This was influenced by the over-weighting of agricultural patches on wider surfaces in *La*, while the narrow, small-sized, elongated agricultural patches in *Fa* were down-weighted by the algorithm. These effects are widened by differences in the overall dimensions of *La* and *Fa*, implying that such area-weighted indices are unsuitable for comparing different scale contexts. Further testing of such indicators in equally-sized contexts would integrate such assumptions [59]. PFD values showed relative low sensitivity to *Fa* and *La* differences (−3% in AGR *Fa* and −1% in TOT *Fa*) compared to the other shape complexity indicators. Its suitability for project purposes should be further checked on wider datasets. Shape complexity in agricultural contexts is suitable for representing the level of patch inter-digitation: high values inform on ecological exchange trends (high exposure to neighboring impacting land uses or high permeability to biotic and resource exchanges with neighboring semi-natural land uses). Thus, their positive or negative ecological value interpretation should be paired with context ecological quality analysis, such as landscape apparatuses and BTC values spatial variability analysis. For instance,

among *Fa*, shape complexity values should be interpreted as positively supporting the NBFS effects as well as the exchanges with the neighboring wooded areas (even if critical exposure to neighboring conventional patches should be considered as well). Moreover, if intended as a measure of land use intensity [69], positive values of shape complexity indices can be indirectly related to heterogeneity and diversity values, especially in rural contexts (checkerboard-shaped landscapes) [55,69,116–118].

Results on landscape composition indices ($DIV_{[1a-1b-2]}$, $DOM_{[1–2]}$, and $LSD_{[1–2]}$) (Tables 2 and 3, Figure 4) highlighted the following traits:

- Among landscape diversity indicators, both $DIV_{[1a-1b-2]}$ showed relative positive contributions of *Fa* to the agricultural sub-system heterogeneity (+86%, +98%, and +124% in AGR *Fa*, respectively). Both $DIV_{[1a-2]}$ showed positive contributions of *Fa* to the total landscape system diversification, while lower values were registered for the *Fa* natural sub-system diversity (both for $DIV_{[1a-1b-2]}$), highlighting the need for further improvement of natural landscape features and their configuration in the pilot case study in order to better counterbalance its positive contributions towards the surrounding context. $DIV_2$ showed the highest relative sensitivity to changes (both for the agricultural subsystem and for the total area), and it should be considered to have higher congruency as it is a normalized version of $DIV_{1a}$ and its values do not depend on the number of land use categories, being more independent of scale [59];
- As expected, $DOM_{[1–2]}$ indicators are inversely correlated to $DIV_{[1a-1b-2]}$ ones (Table 3), and consequently provide redundant information. They were the basis for computing $LSD_{[1–2]}$ values, which showed a positive contribution of *Fa* to the agricultural sub-system diversification [respectively, +41% ($LSD_1$) and +102% ($LSD_2$) in AGR *Fa*]. In comparison, the total landscape system $LSD_{[1–2]}$ showed smaller changes (+2% and +44% in TOT *Fa*, respectively), with $LSD_2$ showing a higher relative sensitivity to changes between total *La* and *Fa*, in line with the trend between $DIV_{1a-2}$ indicators. In line with all $DIV_{[1a-1b-2]}$ values, higher $LSD_{[1–2]}$ values were registered for the *La* on a natural sub-system ($LSD_1$ −51% for NAT *Fa* and $LSD_2$ −37% for NAT *Fa*), confirming the above-cited $DIV_{[1a-1b-2]}$ value interpretation;
- All $DIV_{[1a-1b-2]}$ and $LSD_{[1–2]}$ indicators are positively correlated between each other (Table 3), consequently bringing redundant information, as was expected, and suggesting the opportunity of their screening. Indeed, they were included to allow the comparison of their different performances (a preliminary assessment of their sensitivity to changes among the different studied contexts).

Taken together, landscape composition indices, representing landscape heterogeneity values, showed good relative sensitivity to *La* and *Fa* agricultural and natural sub-system differences, which should be further tested on wider datasets for a comprehensive and coherent sensitivity screening. The dependence of the different indicators on other spatial factors should also be taken into account (first of all, changes in the scale of analysis). Especially, the $DIV_2$ and $LSD_2$ indices were found to be good informative metrics and appeared to be the most suitable ones for fitting the study purposes.

The highlighted positive contribution of *Fa* to AGR and TOT landscape heterogeneity supports the hypothesis of the contribution given by NBFS approaches to agroecosystem diversification, supporting the following related functions among rural land:

- Higher habitat diversification enables source populations to spillover from semi-natural patches to intensively managed patches, balancing generalist and specialist species presences, rising species richness values of different taxa, and reducing local extinction risks [70,102,104,105,111,119–126];
- Activation of buffering functions at the natural-agricultural fringe, moderating the impacts derived from intensive local land use through diversified diachronic microhabitat provisioning (source and refuge areas) [54,103].

As a whole, the results on landscape structural indicators were useful in underlying the contributions given by the pilot farm's NBFS-oriented approach to landscape diversification,

buffering the negative effects on structural and functional biodiversity values derived from the oversimplification patterns that characterise conventional agricultural landscape ecological systems.

### 3.1.2. Landscape Functional Indicators Results

In line with landscape apparatus analysis, BTC values of spatial distribution (Figure 5a,b) highlight the relative high contribution given by woody patches to the overall meta-stability of the studied context, whereas the agricultural matrix shows a significant absence of inter-spread positive BTC values components, except for a central area where the pilot farm is also located. Indeed, the BTC analysis underlined the positive contribution given by the diversified (over space and time scales) land uses among the pilot farm, which represents a *unicum* among the agricultural matrix. This highlights the potential contribution of widespread implementation of NBFS to the release of higher metastability values across rural land outside natural areas, which could positively enhance the homeostatic and homeorhetic capacities of this agricultural system and, consequently, its resistance and adaptation capacities. These statements are aligned with the results obtained on the other structural and functional landscape ecology indices.

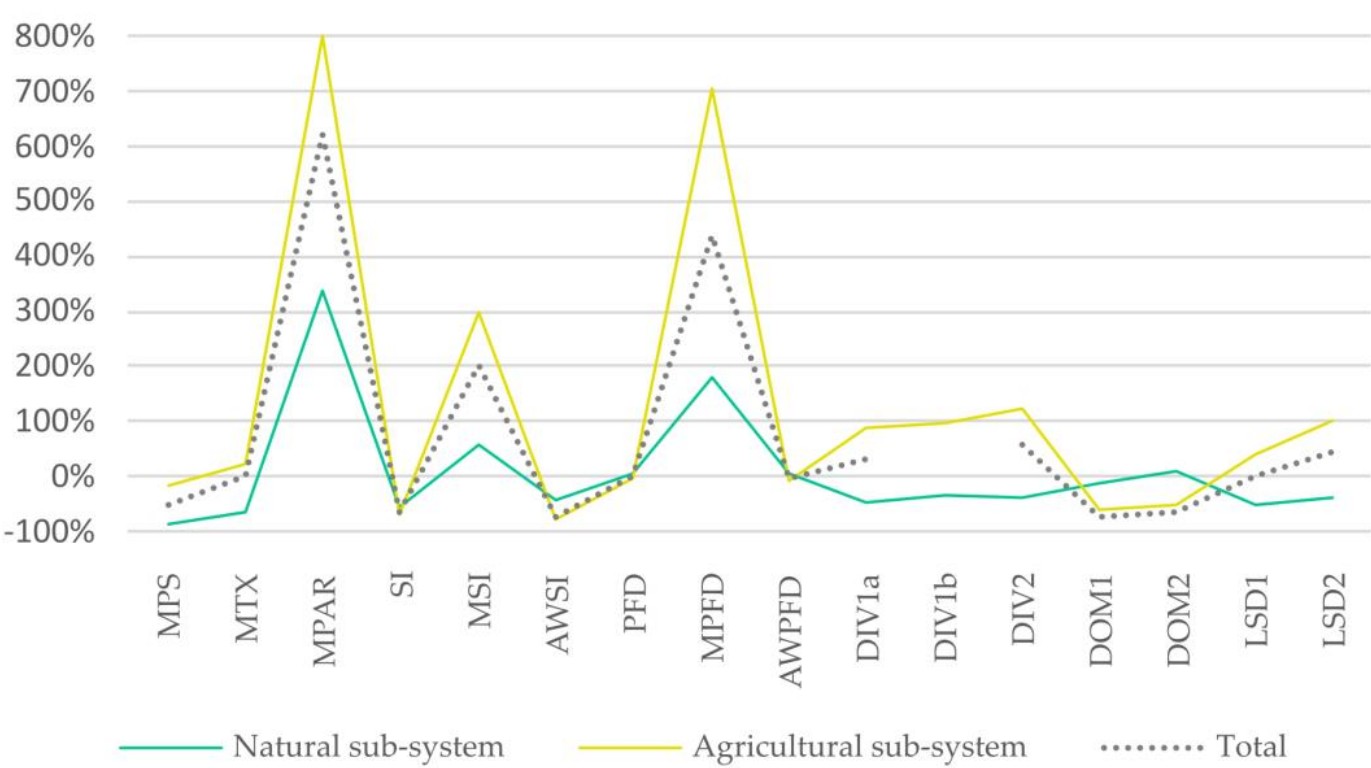

**Figure 4.** Percentage gaps between farm (*Fa*) and local (*La*) scale landscape structural indicator values, calculated separately for the natural and agricultural sub-systems, and then jointly for the total landscape system [Shape Indices (MPS: Medium patches size; MTX: Matrix; MPAR: Mean Perimeter area ratio; SI: Shape index; MSI: Mean Shape index; AWSI: Area weighted mean shape index; PFD: Patch fractal dimension; MPFD: Mean patch fractal dimension; and AWPFD: Area weighted mean patch fractal dimension) and Composition Indices (DIV1a: Diversity_1a/tot; DIV1b: Diversity_1b /landscape element; DIV2: Diversity_2; DOM1: Dominance_1; DOM2: Dominance_2; LSD1: Landscape Structural Diversity_1; and LSD2: Landscape Structural Diversity_2)].

**Table 2.** Results on landscape structural indicators [Shape Indices (NP: number of patches; MPS: Medium patches size; MTX: Matrix; MPAR: Mean Perimeter area ratio; SI: Shape index; MSI: Mean Shape index; AWSI: Area weighted mean shape index; PFD: Patch fractal dimension; MPFD: Mean patch fractal dimension; and AWPFD: Area weighted mean patch fractal dimension) and Composition Indices (DIV1a: Diversity_1a/tot; DIV1b: Diversity_1b/landscape element; DIV2: Diversity_2; DOM1: Dominance_1; DOM2: Dominance_2; LSD1: Landscape Structural Diversity_1; LSD2: Landscape Structural Diversity_2)], comparing local (*La*) and farm (*Fa*) scale levels of analysis, respectively for the natural (NAT), agricultural (AGR), and anthropic (ANT) landscape sub-systems and for the total landscape system (TOT) (in bold), for which the difference between *Fa* and *La* is shown (GAP) (see Appendix A; Table A1 for details on the studied indicators)].

| | Landscape Sbu-system | Area [ha] | Perimeter [m] | NP | MPS | MTX | MPAR | SI | MSI | AWSI | PFD | MPFD | AWPFD | DIV1a | DIV1b | DIV2 | DOM1 | DOM2 | LSD1 | LSD2 |
|---|---|---|---|---|---|---|---|---|---|---|---|---|---|---|---|---|---|---|---|---|
| | | | | | | | | | | | | | | | | | Composition Indices | | | |
| **La** | NAT | 76.6 | 25,587 | 49 | 1.12 | 23.5 | 0.010 | 8.2 | 0.47 | 5.4 | 1.50 | 0.16 | 1.46 | 0.5 | 0.6 | 0.2 | 2.4 | 0.8 | 2.6 | 0.6 |
| | AGR | 226.8 | 108,378 | 273 | 0.53 | 69.6 | 0.002 | 20.3 | 0.16 | 13.8 | 1.58 | 0.05 | 1.56 | 0.8 | 0.8 | 0.3 | 2.0 | 0.7 | 4.3 | 1.1 |
| | ANT | 22.2 | 13,886 | 41 | 0.43 | 6.8 | 0.009 | 8.3 | 0.39 | 6.1 | 1.55 | 0.15 | 1.52 | 0.2 | 0.7 | 0.1 | 2.7 | 0.9 | 1.3 | 0.3 |
| | **TOT** | **325.6** | **147,850** | **363** | **0.67** | **100.0** | **0.004** | **23.1** | **0.23** | **11.3** | **1.59** | **0.07** | **1.53** | **1.5** | | **0.54** | **1.34** | **0.46** | **6.72** | **1.85** |
| **Fa** | NAT | 1.05 | 1315 | 7 | 0.14 | 8.7 | 0.046 | 3.6 | 0.74 | 3.0 | 1.55 | 0.44 | 1.53 | 0.2 | 0.4 | 0.1 | 2.2 | 0.9 | 1.3 | 0.4 |
| | AGR | 10.24 | 7375 | 26 | 0.43 | 85.2 | 0.021 | 6.5 | 0.64 | 2.9 | 1.54 | 0.39 | 1.46 | 1.6 | 1.7 | 0.7 | 0.8 | 0.3 | 6.0 | 2.2 |
| | ANT | 0.73 | 1382 | 8 | 0.09 | 6.1 | 0.047 | 4.6 | 0.80 | 3.3 | 1.63 | 0.40 | 1.60 | 0.2 | 0.7 | 0.1 | 2.2 | 0.9 | 1.1 | 0.3 |
| | **TOT** | **12.02** | **10,071** | **41** | **0.32** | **100.0** | **0.031** | **8.2** | **0.68** | **3.0** | **1.58** | **0.40** | **1.47** | **2.0** | | **0.84** | **0.37** | **0.16** | **6.83** | **2.66** |
| **GAP (Fa−La)** | **TOT** | | | | **−0.36** | | **0.026** | **−14.9** | **0.46** | **−8.29** | **−0.01** | **0.33** | **−0.06** | **0.48** | | **0.31** | **−0.97** | **−0.31** | **0.11** | **0.81** |
| | NAT% | | | | −88% | −63% | 338% | −56% | 57% | −45% | 4% | 181% | 5% | −48% | −35% | −38% | −11% | 7% | −51% | −37% |
| | AGR% | | | | −19% | 22% | 801% | −68% | 299% | −79% | −3% | 704% | −6% | 86% | 98% | 124% | −59% | −51% | 41% | 102% |
| | TOT% | | | | **−53%** | **0%** | **623%** | **−65%** | **201%** | **−74%** | **−1%** | **438%** | **−4%** | **31%** | | **58%** | **−72%** | **−66%** | **2%** | **44%** |

**Table 3.** Correlation coefficients matrixes for shape indices (MPAR: Mean Perimeter area ratio; SI: Shape index; MSI: Mean Shape index; AWSI: Area weighted mean shape index; PFD: Patch fractal dimension; MPFD: Mean patch fractal dimension; and AWPFD: Area weighted mean patch fractal dimension) and for composition indices (DIV1a: Diversity_1a /tot; DIV1b: Diversity_1b /landscape element; DIV2: Diversity_2; DOM1: Dominance_1; DOM2: Dominance_2; LSD1: Landscape Structural Diversity_1; and LSD2: Landscape Structural Diversity_2).

| | **Shape Indices** | | | | | | | | **Composition Indices** | | | | | | |
|---|---|---|---|---|---|---|---|---|---|---|---|---|---|---|---|
| | **MPAR** | **SI** | **MSI** | **AWSI** | **PFD** | **MPFD** | **AWPFD** | | **DIV1a** | **DIV1b** | **DIV2** | **DOM1** | **DOM2** | **LSD1** | **LSD2** |
| MPAR | 1.00 | | | | | | | DIV1a | 1.00 | | | | | | |
| SI | −0.76 | 1.00 | | | | | | DIV1b | 0.94 | 1.00 | | | | | |
| MSI | 0.93 | −0.89 | 1.00 | | | | | DIV2 | 0.99 | 0.96 | 1.00 | | | | |
| AWSI | −0.76 | 0.94 | −0.94 | 1.00 | | | | DOM1 | −0.95 | −0.89 | −0.97 | 1.00 | | | |
| PFD | 0.31 | 0.23 | 0.08 | 0.20 | 1.00 | | | DOM2 | −0.99 | −0.96 | −1.00 | 0.97 | 1.00 | | |
| MPFD | 0.91 | −0.82 | 0.96 | −0.89 | 0.12 | 1.00 | | LSD1 | 0.98 | 0.89 | 0.95 | −0.88 | −0.95 | 1.00 | |
| AWPFD | 0.26 | 0.19 | −0.06 | 0.33 | 0.80 | −0.09 | 1.00 | LSD2 | 1.00 | 0.95 | 1.00 | −0.97 | −1.00 | 0.96 | 1.00 |

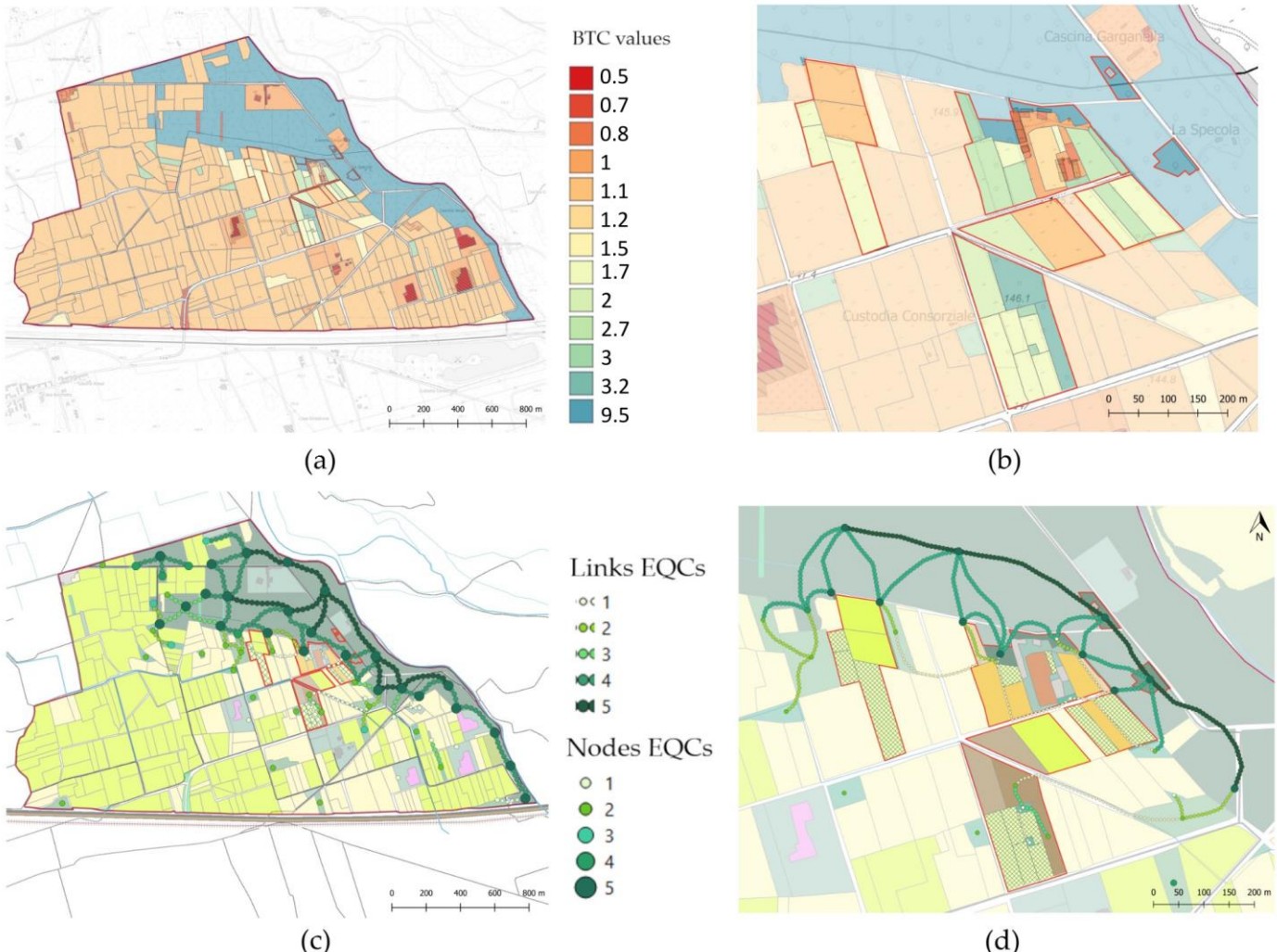

**Figure 5.** (**a**,**b**) Spatial distribution of BTC values related to the land use patch categories at local (**a**) and farm (**b**) scales; (**c**,**d**) spatial configuration of links and nodes at local (**c**) and farm (**d**) scales, showing their belonging to the different Ecological Quality Classes (EQCs).

The analysis of functional connectivity (CON and WCON) and circuitry (CIR and WCIR) indicators (Figure 5c,d) showed good relative sensitivity, with parallel trends between the *Fa* and *La* scales. Results confirm the role of the local scale northeastern woody patches in sustaining the biotic and abiotic fluxes across the landscape system, whereas the low NBFS equipment of the agricultural matrix significantly lowers the overall local scale connectivity and circuitry values. The comparison of the CON and CIR indices between the *La* and *Fa* scales (Figure 6a) highlights the positive contribution on ecological fluxes given by NBFS implementation among the pilot farm (+0.13 on CON; +0.18 on CIR). Indeed, connectivity functions are tightly related to population and metapopulation dynamics and biodiversity values [13,54,113,127–133]. The indices' weighted variants (WCON and WCIR) allowed us to efficiently take into account the role of the effective current ecological status of connectivity components (links), highlighting the distance from optimal conditions and counterbalancing the negative connectivity effects related to generalist species' predisposition to low EQC corridors [134,135]. Indeed, the WCON and WCIR index values show smaller discrepancies between *La* and *Fa* scales (+ 0.06 on the *Fa* WCON and + 0.08 on the *Fa* WCIR, respectively), a result related to the current medium to low ecological quality of the linear landscape features among the pilot farm, which, despite being relatively spread, could benefit from further ecological requalification. In line with this, the distribution of *La* and *Fa* scale links among the different $EQC_y$ (Figure 6b) shows mixed patterns, with low-$EQC_y$ links being more represented among *Fa* scale (+7% on the 1st EQC and +4% on the 2nd EQC) and a higher percentage of the maximum $EQC_y$ at *La* scale (+6% on the 5th EQC), where connectivity functions are mostly expressed by high ecological quality woody patches. These results suggest a good suitability of the EQC synthetic weighting system for monitoring slight changes over time.

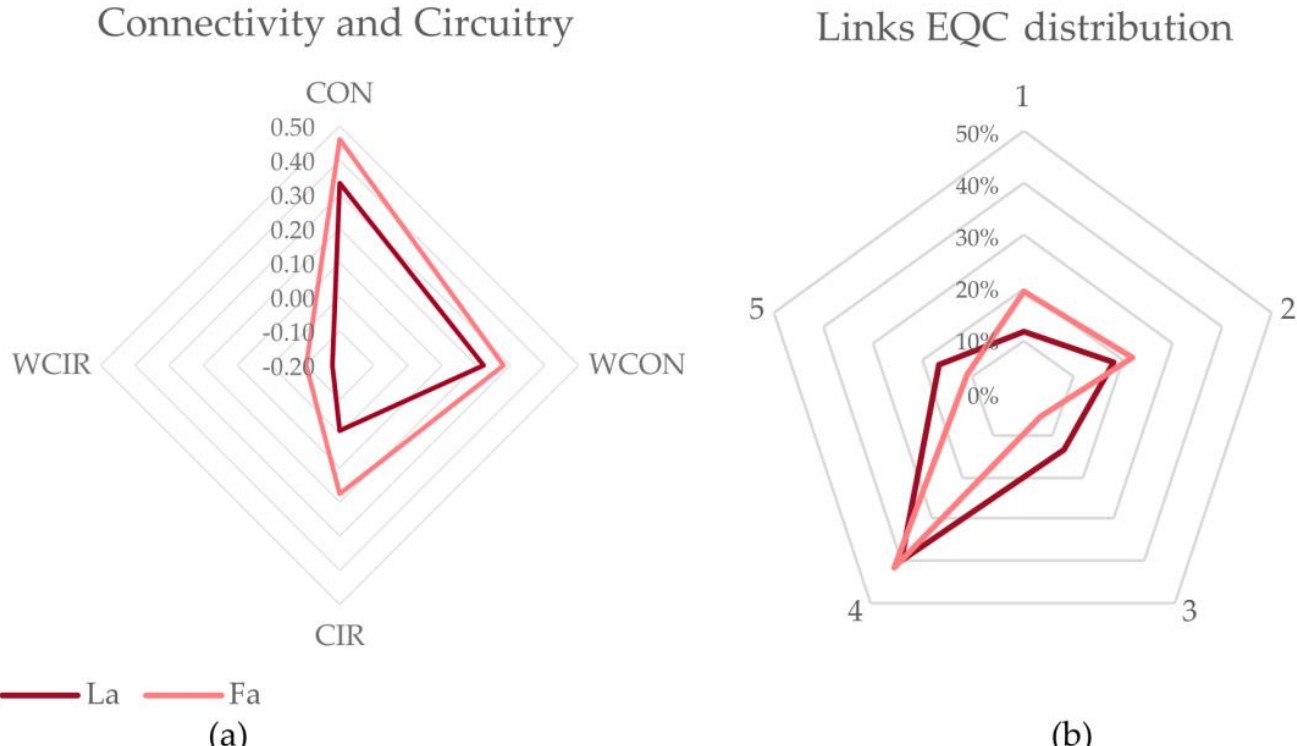

**Figure 6.** (**a**) Comparison of results on connectivity (CON: connectivity and WCON: weighted connectivity) and circuitry (CIR: circuitry and WCIR: weighted circuitry) indices for local (*La*) and farm (*Fa*) scales; and (**b**) comparison of the distribution of *La* and *Fa* scale links among the five $EQC_y$.

In line with these results, the landscape apparatus analysis revealed low connectivity functions throughout the overall *La* context. Their spread across the agricultural matrix

would be pivotal in any corrective management strategy, as was highlighted through the extra-local and local VR analyses.

The local scale VR analysis (Figure 7), as a conceptual result of multiple territorial information overlays and landscape ecology analyses, gives a synthetic qualitative outline of the predominant spatial configuration of ecological fluxes of information, genetic, trophic, and biotic resources across the *La* scale area. Such an analysis shows the positive current and potential contribution that the pilot farm under study can make in relation to its surrounding context. The current synthesis helps in prioritizing the featuring and allocation of the ecological reconfiguring strategies as a response to the criticalities highlighted by the landscape ecology analyses, which are currently impacting floristic-vegetational ecological quality and agrobiodiversity values (low connectivity and circuitry levels among the agricultural matrix, over-sized patches acting as source areas, relative low spatial heterogeneity, barrier effect derived from major infrastructures) (see Sections 3.1.1 and 3.1.2). The strategic mending and buffering role of the agricultural areas neighboring the wooded areas (where the pilot farm is also located) are highlighted.

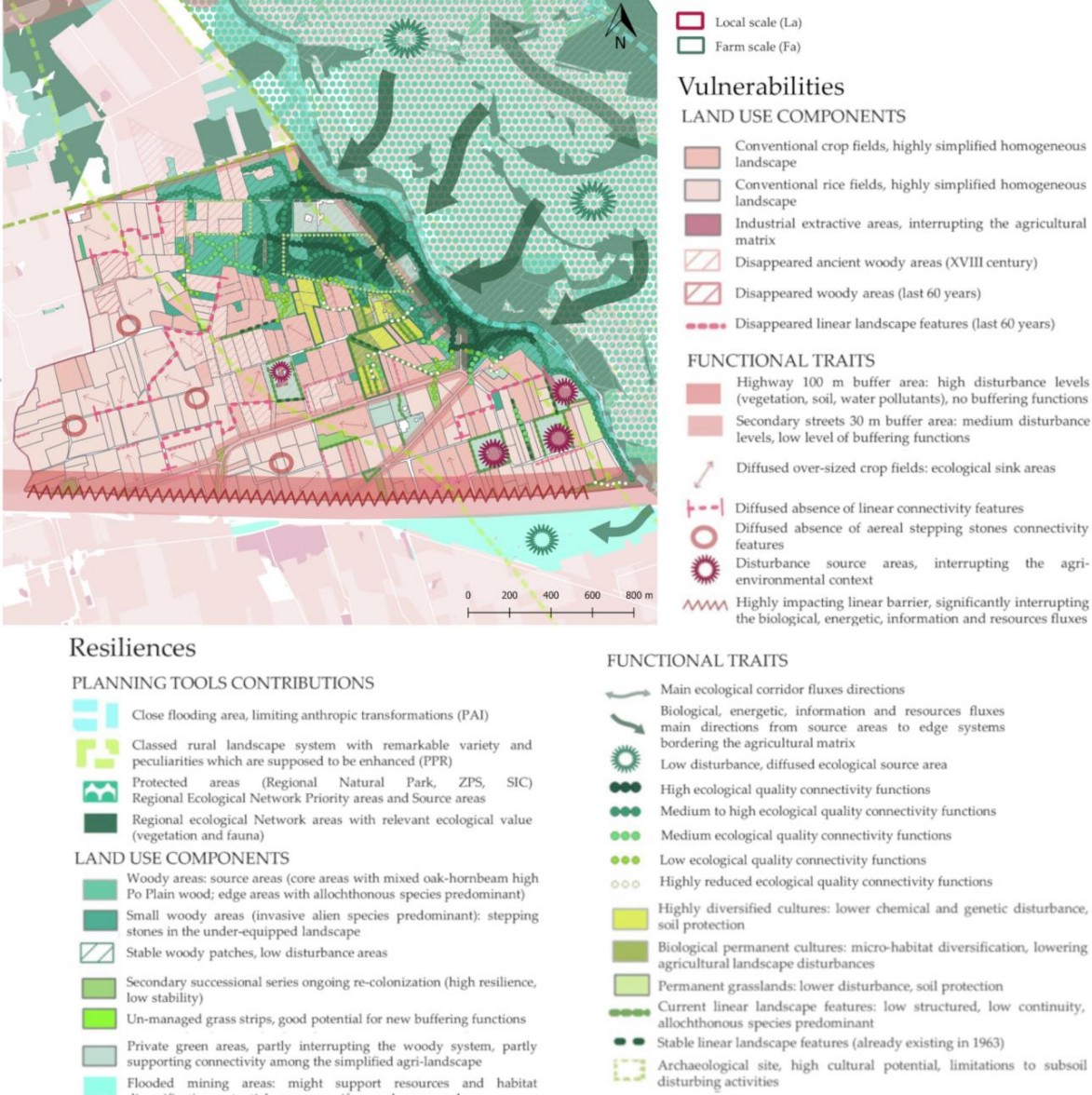

**Figure 7.** Local scale VR analysis.

## 4. Conclusions

The results presented in this study were based on a workflow that was intended to examine different facets of the ecological functions and processes that interact among agroecosystems. The study allows the preliminary selection and screening of the most sensitive and informative analyses to be integrated into an NBFS assessment methodology in which analytical steps are closely interlinked with management practice targeting and farmland ecological design strategic orientation. Multi-scale evaluations, with different levels of detail, were useful for representing the inter-linkages existing between farm-scale ecological traits and the surrounding local landscape context, which is influenced by and influences the surrounding extra-local landscape system ecological configuration (more detailed analyses).

The results of the *Fa* and *La* scale levels of analysis allow for the identification of the main ecological traits associated with the pilot farm NBFS management, demonstrating its positive contributions when compared to the business as usual conventionally managed surrounding *La* scale context. The ecological interpretation is based on the acknowledged role of landscape metrics as surrogates for agrobiodiversity values. Nonetheless, the in-view comparison of landscape metrics results with multi-scale floristic-vegetational indicators assessment would broaden the ecological interpretation, allowing an examination of the use of landscape metrics as correlates of agrobiodiversity indices, screening and validating their value as context-specific agrobiodiversity proxies.

From a methodological point of view, this study helps in preliminarily identifying a set of indicators suitable for the multi-scale assessment of NBFS contributions to agrobiodiversity in the studied pedo-climatic context. High sensitivity to different management options, low redundancy, and good suitability for multi-scale comparisons were the guiding evaluation criteria. Further research on larger datasets derived from other pilot case studies with similar pedo-climatic conditions is required to improve the statistical consistency of the methodology presented here. In particular, the comparison of pilot case studies with conventional farms at the same scale (*Fa*) would allow a finer screening of the most suitable indicators, integrating the preliminary results better by considering the sensitivity of the indicators to changes in scales of analysis.

Overall, the results presented in this study are expected to provide new useful tools that can be applied to farming systems belonging to similar pedo-climatic conditions and integrate current knowledge on synthetic agrobiodiversity indicators.

Moreover, this multi-scale approach appeared to be suitable for interlinking best-practice assessments (a result-based monitoring approach) with effective best-practice promotion and spread (a science-driven strategic NBFS design approach). Indeed, current ecosystem functions supported by the NBFS were interlinked with potential future enhancements of ecosystem functions through VR analysis. Such an approach paves the way for integrating the ecosystem functions and services approach into the assessment and design processes, both on a spatial-qualitative and quantitative basis [136–141].

**Supplementary Materials:** The following supporting information can be downloaded at: https://www.mdpi.com/article/10.3390/agronomy13010233/s1, Table S1: Local scale (La)—patches; Table S2: Local scale (La)—indexes (1); Table S3: Local scale (La)—indexes (2); Table S4: Local scale (La)—Links EQCyi; Table S5: Local scale (La)—Connectivity and circuitry; Table S6: Farm scale (Fa)—patches; Table S7: Farm scale (Fa)—indexes (1); Table S8: Farm scale (Fa)—indexes (2); Table S9: Farm scale (Fa)—Links EQCyi; Table S10: Farm scale (Fa)—Connectivity and circuitry.

**Author Contributions:** Conceptualization, I.V. and G.C.; methodology, I.V.; software, G.C.; validation, I.V.; formal analysis, G.C.; investigation, I.V. and G.C.; resources, I.V.; data curation, I.V. and G.C.; writing—original draft preparation, G.C.; writing—review and editing, I.V.; visualization, I.V. and G.C.; supervision, I.V.; project administration, I.V.; funding acquisition, I.V. All authors have read and agreed to the published version of the manuscript.

**Funding:** This research received no external funding.

**Data Availability Statement:** The data presented in this study are available on request from the corresponding author.

**Conflicts of Interest:** The authors declare no conflict of interest.

## Appendix A

**Table A1.** Set of indicators under study.

| | Indicators | Equation | References |
|---|---|---|---|
| **PATCHES METRICS** | Medium patches size (MPS) | $MPS = \frac{\sum_{i=1}^{n} A_i / N_i}{LU}$<br>$A_i$ = total area of each land use categories patches<br>$N_i$ = no. of patches for each land use categories<br>$LU$ = no. of land use categories | [59] |
| | Matrix (MTX) | $MTX = \frac{\sum_{i=1}^{n} A_i \times 100}{A_{tot}}$<br>$A_i$ = total area of each land use categories patches<br>$A_{tot}$ = total area | [59] |
| **SHAPE INDICES** | Mean Perimeter area ratio (MPAR) | $MPAR = \frac{\sum_{i=1}^{n} PAR_i}{NP}$<br>$PAR_i = \frac{P_i}{A_i}$<br>$P_i$ = Perimeter of each land use category patches<br>$NP$ = total no. of patches | [69] |
| | Shape index (SI) | $SI = \frac{P_i}{2 \times \sqrt{\pi \times A_i}}$ | [69] |
| | Mean Shape Index (MSI) | $MSI = \frac{\sum_{i=1}^{n} SI}{NP}$ | [69] |
| | Area weighted mean shape index (AWSI) | $AWSI = \frac{\sum_{i=1}^{n} SI \times A_i}{\sum_{i=1}^{n} A_i}$ | [69] |
| | Patch fractal dimension (PFD) | $PFD = \frac{2 \times \ln P_i}{\ln A_i}$ | [59,69] |
| | Mean patch fractal dimension (MPFD) | $MPFD = \frac{\sum_{i=1}^{n} PFD_i}{NP}$ | [69] |
| | Area weighted mean patch fractal dimension (AWPFD) | $AWPFD = \frac{\sum_{i=1}^{n} PFD_i \times A_i}{\sum_{i=1}^{n} A_i}$ | [69] |
| **COMPOSITION INDICES** | Diversity_1a/tot (DIV1a) | $DIV_{1a} = -\sum_{i=1}^{n} \frac{A_i}{A_{tot}} \times \ln \frac{A_i}{A_{tot}}$ | [59] |
| | Diversity_1b/landscape element (DIV1b) | $DIV_{1b} = -\sum_{i=1}^{n} \frac{A_i}{A_y} \times \ln \frac{A_i}{A_y}$<br>$A_y$ = total area of each landscape system (natural, agricultural, and anthropic) | [59] |
| | Diversity_2 (DIV2) | $DIV_2 = \frac{-\sum_{i=1}^{n} \frac{A_i}{A_{tot}} \times \ln \frac{A_i}{A_{tot}}}{\ln S}$<br>$\ln S = \max(DIV)$<br>$S$ = no. of land use categories | [59,142] |
| | Dominance_1 (DOM1) | $DOM_1 = \ln S + \sum_{i=1}^{n} \frac{A_i}{A_{tot}} \times \ln \frac{A_i}{A_{tot}}$ | [82,84] |
| | Dominance_2 (DOM2) | $DOM_2 = \ln S + \frac{\sum_{i=1}^{n} \frac{A_i}{A_{tot}} \times \ln \frac{A_i}{A_{tot}}}{\ln S}$ | [59] |
| | Landscape Structural Diversity_1 (LSD1) | $LSD_1 = DIV_1 \times (3 + DOM_1)$ | [82,84] |
| | Landscape Structural Diversity_2 (LSD2) | $LSD_2 = DIV_2 \times (3 + DOM_2)$ | [82,84] |

**Table A1.** *Cont.*

| | Indicators | Equation | References |
|---|---|---|---|
| **CONNECTIVITY INDICES** | Connectivity (CON) | $CON = \frac{L}{[3 \times (N-2)]}$<br>$L$ = no. of links<br>$N$ = no. of nodes | [95] |
| | Weighted connectivity (WCON) | $WCON = \frac{\sum_{i=1}^{5} L_i \times W_i}{[3 \times (N-2)]}$<br>$L_i$ = no. of links for each Ecological Quality<br>Class ($EQC_i$ = [1–5])<br>$W_i = EQC_i$ weight:<br>$W_i = \frac{EQC_i}{EQC_{max}}$ | [95]<br>modified<br>by authors |
| | Circuitry (CIR) | $CIR = \frac{(L-N+1)}{[2 \times (N-5)]}$ | [95] |
| | Weighted circuitry (WCIR) | $WCIR = \frac{[(\sum_{i=1}^{5} L_i \times W_i) - N + 1)]}{[2 \times (N-5)]}$<br>$L_i$ = no. of links for each Ecological Quality<br>Class ($EQC_i$ = [1–5])<br>$W_i = EQC_i$ weight (as above) | [95]<br>modified<br>by authors |
| **FUNCTIONALITY INDICES** | Biological Territorial Capacity (BTC) | $BTC = \sum_{i=1}^{n} BTC_i \times A_i$<br>$BTC_i$ = BTC value attributed to each land<br>use category (tabulated values;<br>see references) | [73,94] |

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
