# Peer review of "Validating the Contribution of Nature-Based Farming Solutions (NBFS) to Agrobiodiversity Values through a Multi-Scale Landscape Approach"

_agronomy, doi:10.3390/agronomy13010233_

Round 1

Reviewer 1 Report

Validating NBFS contributions to agrobiodiversity values through a multi-scale landscape approach

Line 17: were led at over-local; please consider rewording sentence e.g. were undertaken at…..

Line 31: as landscape features management; please consider rewording sentence i.e. another word is needed before management?

Line 47: performance (singular)

Line 52: indicator (singular)

Line 55: us to use them

Overall, the introduction has many long sentences and some repletion of key points / terms. I would encourage the authors to shorten sentences and review the text for repetition. For examples, lines 74 and 75 are a repetition of 63 and 64 i.e. the point has already been made.

Line 76 their in-plan; who are “their”?

Line 90 indicator not indicators

Line 89-93 Very long sentence that makes it difficult for the reader to follow the content

Line 144: Würm

Line 146: production not productions

Line 169: of 132 taxa

Line 188-193: Sentence very long; please revise

Are you able to define over (extra) local and local scales in km2? it would be useful to clear how you are defining over local

Line 246: They also resulted; Instead They were also positively correlated

Line 305: an coherent sensitivity screening; reword sentence it is unclear

The text needs to be reviewed by someone whose English is their first language because the meaning of many sentences is obscured by overly complex and unclear grammar. Not in every aspect of the paper but it is sufficiently high to recommend that help with the English grammar is required, which will significantly improve what is a very interesting study. I have only mentioned a few instances above but the entire paper will need to be reviewed.

Overall, the paper is worthy of publication. It presents a detailed methodology for the study of landscape ecology and agrobiodiversity. It is an extremely detailed study with very good figures and data tables. It is written in a very concise manner, but as noted above would benefit from assistance with the English grammar.

Author Response

Thank you for the careful and timely review. We have taken note of all suggestions and made all corrections. 

Reviewer 2 Report

Introduction: The introduction is very concise, clearly presented and supported with good references. P. 35. The ref. is already mentioned by number, there is no need to repeat it.

Results and discussion

p.191-203. Probably it is too premature to put this par. at the very beginning because it creates a lot of questions. These results are basically supported by the following results and figures. As such a better restructuring of this text and figures is needed.

Conclusions

Please mention any future research implications.

General comments:

It is a very interesting, well written and presented article and it can be a very valuable input to the agricultural research. However, the authors need to proceed with these minor changes in order for the article to be published.

Author Response

Thank you for the review, we have noted the suggestions and modified the points indicated.